# Green Aspects in Molecularly Imprinted Polymers by Biomass Waste Utilization

**DOI:** 10.3390/polym13152430

**Published:** 2021-07-23

**Authors:** Roberta Del Sole, Giuseppe Mele, Ermelinda Bloise, Lucia Mergola

**Affiliations:** Department of Engineering for Innovation, University of Salento, via per Monteroni Km1, 73100 Lecce, Italy; giuseppe.mele@unisalento.it (G.M.); ermelinda.bloise@unisalento.it (E.B.); lucia.mergola@unisalento.it (L.M.)

**Keywords:** biomass waste, molecularly imprinted polymers, ion imprinted polymers, chitosan, cellulose, cyclodextrin, carbon dots, activated carbon, magnetic polymers, surface imprinting

## Abstract

Molecular Imprinting Polymer (MIP) technology is a technique to design artificial receptors with a predetermined selectivity and specificity for a given analyte, which can be used as ideal materials in various application fields. In the last decades, MIP technology has gained much attention from the scientific world as summarized in several reviews with this topic. Furthermore, green synthesis in chemistry is nowadays one of the essential aspects to be taken into consideration in the development of novel products. In accordance with this feature, the MIP community more recently devoted considerable research and development efforts on eco-friendly processes. Among other materials, biomass waste, which is a big environmental problem because most of it is discarded, can represent a potential sustainable alternative source in green synthesis, which can be addressed to the production of high-value carbon-based materials with different applications. This review aims to focus and explore in detail the recent progress in the use of biomass waste for imprinted polymers preparation. Specifically, different types of biomass waste in MIP preparation will be exploited: chitosan, cellulose, activated carbon, carbon dots, cyclodextrins, and waste extracts, describing the approaches used in the synthesis of MIPs combined with biomass waste derivatives.

## 1. Introduction

### 1.1. The Value of Biomass Waste

The aim of green chemistry can be briefly described with a working definition in the following sentence: green chemistry efficiently utilizes (preferably renewable) raw materials, eliminates waste, and avoids the use of toxic and/or hazardous reagents and solvents in the manufacture and application of chemical products. Nowadays, its concepts are accepted in academic and industrial spheres on a worldwide basis, recognizing that the implementation of green chemistry strategies is beneficial both to lead to a cleaner and more sustainable world but that it is also economically advantageous with many positive social effects [1,2,3,4,5].

One of the aspects of green and sustainable chemistry that will be analyzed in this review is the valorization of biomass waste [6,7,8,9,10,11,12]. Currently, the amount of biomass waste generated from different sources has been dramatically increased.

In the light of these considerations, one of the main objectives is to drastically reduce the environmental impact of waste with the scope to promote the concept for their recycling and transforming it into value added products.

Generally, biomass waste has been addressed for power generation purposes and the production of valuable carbon-based compounds; however, it is important to remark that, in some cases, waste biomass-assisted synthesis is a less costly, more environmentally friendly, and renewable strategy, and therefore wastes may become ideal renewable resources for production of functionally engineered macro or nanomaterials.

Numerous physical, chemical, and biological routes have been investigated to optimize the utilization of biomass waste with a special attention for green and sustainable techniques.

Biomass waste is an environmentally friendly renewable resource that can be obtained from agricultural residues, human activity waste, fishery, wood industry, livestock waste, and so on. Biomass waste is a widespread natural carbon source which mainly contains cellulose, hemicellulose, lignin, chitin, ash, and proteins, and thus it is particularly suitable to be used as precursor to prepare high valued carbon-based materials stimulating a sustainable approach. Recently, carbon materials obtained from biomass waste have shown potential applications in hydrogen storage, biomedicine, sorption materials, and so on. However even if biomass waste has a high carbon amount, for instance, plant biomass waste has values of cellulose of 30–60%, hemicellulose of 20–40%, and lignin of 15–25%, only a small amount of biomass waste is utilized now, while most of it is discarded [13,14].

### 1.2. Fundaments of MIPs

At present, green chemistry principles are fundamental in many scientific fields with an enhanced emphasis in chemical processes, and MIPs cannot be excluded from this focus.

Molecularly imprinted polymers (MIPs) are based on the formation of specific interactions between a template (atom, ion, molecule, complex or a molecular, ionic, or macromolecular assembly, including microorganisms) and a functional monomer and the successive polymerization in the presence of a large excess of a crosslinking agent. Then, the template is removed from the crosslinked polymer leaving into the polymeric network specific recognition sites complementary in shape, size, and chemical functionality to the template molecule. Thus, the resultant polymer can bind specifically the template molecules. The main advantages of MIPs are their high selectivity and affinity for the target molecule used in the imprinting procedure similarly to the well-known lock and key model used in biological processes. Imprinted polymers, compared to biological systems such as proteins and nucleic acids, have higher physical robustness, strength, and resistance to elevated temperatures and pressures and inertness towards acids, bases, metal ions, and organic solvents. In addition, their production is less expensive with a very high storage life, keeping their recognition capacity also for several years at room temperature [15,16,17,18]. MIPs’ advantageous features permitted an extensive application in analytical field such as foods, drugs, biological, and environmental sectors for the detection, preconcentration or separation of several analytes [19,20,21,22]. Other MIPs applications such as drug delivery [23] and catalysis [24,25,26], tissue engineering [27] have be also considered. It is worth noting that several different MIPs formats have been developed such as bulk or monoliths, microspheres and core–shell materials, magnetically susceptible and stir-bar imprinted materials which are applicable as sorbents of solid-phase extraction. MIP sorbents can determine target analytes and ions in a very complex environment such as blood, urine, soil, or food, and they have gained interest in trace analysis of pollutants in environmental samples. On the other side, MIPs in the sensor field have seen electrochemical sensing as the preferred analytical technique, followed by optical detection with high improvements in sensitivity as well as a shift in research and development toward real-life applications and point-of-care testing in real human samples. Moreover, the research in MIP sensor devices is nowadays moving from laboratory research to large-scale manufacturing [22].

In the last twenty years, the number of articles and reviews on the topic of MIPs has continued to increase, as can be seen from a research on Scopus platform (Figure 1), which confirms the large interest of the scientific community towards this area that allowed continuous advances in design, preparation, characterization, and application of MIPs.

Remarkable results have been achieved in the molecular imprinting field making it a mature technology even if there are significant findings and opportunities still open. Some work on MIPs takes into consideration green concepts, while others apply green principles without point out them. For instance, MIPs have a very promising future with great potential development in the food industry, theranostics and pharmaceutics but their applications in these fields, where toxicity specially needs to be contained, require more efforts to engage green approach to produce “green” MIPs. In the last decade, several high-quality reviews have been written on molecular imprinting [15,16,17,18], but most of them have dealt with the fundamental aspects and characteristic applications of MIPs while few review articles on novel techniques related to their preparation. Recently, only a few reviews highlighting the green aspect in MIPs preparation have been published [20,28,29,30,31]. On the other hand, to the best of our knowledge, there are no reviews focused on the transformation of biomass waste to produce natural substrates that can be used in place of other toxic substances generally employed for MIPs design.

Herein, we will explore the recent findings in the use of biomass wastes, such as natural polymers, as backbone-based materials to prepare green and biodegradable MIPs. This review aims to provide an overview on the value of using biomass waste as a key element in MIP preparation describing the different possible approaches in MIPs synthesis. In particular, several high value biomass waste derivatives will be considered in combination with MIPs.

The aim of this review is to give to the reader an overview of recent works that have seen the use of biomass in the preparation of MIPs. The next paragraph has been divided into six subparagraphs each of them dealing with a specific biomass waste. In detail, the following wastes will be taken into consideration: chitosan, cellulose, activated carbon, carbon dots, cyclodextrins, and waste extracts. The herein paper is not a critical collection of results of the above-cited topics but a comprehensive and useful summary of the state-of-art as starting point for future development in the topic of biomass waste for MIPs preparation.

## 2. Different Biomass Waste in MIP Technology

In the following subparagraphs, the added value of each different class of biomass waste in MIPs design will be highlighted. Their properties and preparation methods and some relevant examples on their utilization in MIP field will be shown.

### 2.1. Chitosan

It is well known that Chitosan (CS) is the second most abundant natural biopolymer after cellulose which can be easily obtained from chitin, a widespread fish biomass waste. Chitin is the main structural component in the exoskeleton of various marine invertebrates. The natural polysaccharide CS has been attracting great attention in the scientific world, in fact the research of the descriptor “chitosan” on Scopus platform showed a regular exponential increase of papers in the last 30 years that has moved from only 120 papers in 1991 to 8183 papers in 2020. The big interest on chitosan material can be justified from two points of view. First, CS satisfies various requests of green chemistry thanks to its abundance, nontoxicity, biodegradability, and biocompatibility. Moreover, from the second point of view, it has interesting physicochemical and biological properties such as water compatibility, the presence of chemical functional groups such as free amino and hydroxyl groups along its backbone capable of making interaction with metal ions and organic molecules or capable of making chemical reactions such as crosslinking or grafting processes, but also antimicrobial and antibacterial activities. Thus, CS has found successful utilization in several fields ranging from biomedical application (drug delivery, tissue regeneration, and wound healing), to wastewater treatment (adsorbent and sensors) till food, agriculture, cosmetics, papermaking, and textile applications. For instance, note that in the field of metal ion recovery technology, high importance is devoted in the use of green adsorption approaches such as the use of bioderived materials and chitosan-based adsorbents that represent an ideal example since the amine and hydroxyl groups of the CS chain can react with metal ions through chelation or ion-exchange mechanisms [32,33,34,35,36].

Chitin is a linear polysaccharide, made up of (1,4)-linked *N*-acetyl-d-glucosamine units, which is not used as is, but it is often converted into chitosan. Chitosan is produced by the alkaline partial deacetylation of chitin through enzymatic or chemical processes. The degree of deacetylation (DD), which determines the amount of free amino groups into the biopolymer chain, and it is responsible of its solubility and acid–base behavior, allows also to distinguish between chitin and chitosan: when DD values are lower than 60% the polymer is named chitin while for a DD value higher than 60% the polymer is named chitosan [37,38]. 

Chitosan mainly consists of β(1-4)-2amino-2-deoxy-d-glucose units with three types of free functional groups in the backbone: one amine group and two primary and secondary hydroxyl groups. Thus, CS structure can undergo numerous chemical reactions such as the reaction with aldehydes and acetones, hydrogen bonds, crosslinking, grafting, etc. When CS is dissolved in an acidic solution, amine groups protonate, and the polymer becomes positively charged and can interact with anionic molecules. CS behavior can be affected by several parameters such as pH and humidity. CS solubility in an acidic water environment allows to obtain CS in various forms, such as films, nanofibers, hydrogels, beads, membranes, or pastes. On the other hand, in some applications, such as the removal and separation of metal ions or organic pollutants from water solutions, adsorbents should have insolubility in water and also good mechanical strength in acid or alkali solutions. Thus, treatments of chitosan with crosslinking agents or by grafting are common strategies to improve its mechanical stability and insolubility.

The functional properties of CS can be also improved when combined with other materials. Therefore, starting from the natural capability of chelating metal ions and from the numerous excellent properties described above, chitosan is nowadays considered an optimal functional monomer for ion imprinted polymers (IIPs) or molecularly imprinted polymers making it a promising alternative to other functional monomers. In the last decades, many researchers have synthesized several chitosan-based IIPs using chitosan as functional monomer or supporting matrix [37,38].

The first paper with a clear application of CS in IIP technology dates to 2001. Tianwei at al. demonstrated that a Ni(II)-imprinted chitosan resin, obtained by crosslinking with epichlorohydrin and ethylene glycol diglycidyl ether, had a good chemical and physical stability and could be used many times without losing adsorption capacity, considerably enhancing the adsorption capacity and the selectivity toward metal ions [39]. From then to now, tremendous progress has been made as it has been summarized in some interesting reviews with this focus [40,41,42]. Xu et al. [40] wrote a review on the use of chitosan in MIPs in which described the commonly used crosslinkers agents, highlighting the crosslinking mechanisms. In 2020, two other reviews appeared dealing with CS-based MIPs with the aims to provide an overview of the value to use chitosan as a functional monomer with a focus on its applications in electrochemical sensors [41], while Karrat et al. presented a brief overview of the recent applications of MIPs and IIPs composites based on chitosan with the focus on separation and sensing applications [42].

CS has been used primarily as functional monomer or support material to obtain IIPs or MIPs for the recognition of many metal ions, such as heavy metal for environmental application, rare earths, and precious metals, but also for the recognition of organic molecules, such as organic pollutants, protein, and chiral compounds. They were achieved in a multitude of forms ranging from membranes, beads, resins, fibers, core–shell structures, foams to xerogels.

Even if chitosan satisfies green principles because it is a widespread and cheap fish biomass waste, the green principles are not always applied in the whole process that use chitosan. Thus, in a more comprehensive green perspective, it is desirable that researchers keep in mind and evaluate the green principles of all the design steps.

In the design of eco-friendly chitosan-based IIPs or MIPs nontoxic crosslinkers, nontoxic initiators and nontoxic solvents have been considered thus extending the application of MIP in pharmaceutical and food fields too. Herein only some recent works will be described, choosing some sophisticated design, in which different forms or types of CS-based MIPs or IIPs composites have been realized with the aim to show the great flexibility in the utilization of CS in MIPs field.

In many studies CS was used as imprinting functional polymer that, via crosslinking reactions in presence of the template, creates the imprinting effect and enhances insolubility and mechanical strength. Various crosslinker agents have been used such as glutaraldehyde [43,44] epichlorohydrin [45,46], sulfuric acid, glyoxal [47], and so on.

Moreover, recently an increasing number of examples of MIPs or IIPs combined systems, in which chitosan has been combined with other materials, mainly nanomaterials, has been observed. In these contexts, we can cite the use of magnetic nanoparticles, graphene, multiwalled carbon nanotubes (MWCNTs), gold nanoparticles (AuNPs) capable of adding other optimal properties such as magnetic, electrical or high surface area [42].

Bagheri et al. [48,49] realized two different core–shell CS-based magnetic MIPs combining the utilization of a green synthesis strategy and the versatile magnetic solid-phase extraction (MSPE) technique, with the advantage of an external magnet that avoids filtration, centrifuge and precipitation procedures. Fe_3_O_4_ nanoparticles were used as core and CS as shell. In his first work [48], a dummy MIP was prepared in aqueous solution and used as sorbents of MSPE for an efficient clean-up and pre-concentration acrylamide (AA) in biscuit samples. First, Fe_3_O_4_ nanoparticles were synthesized by coprecipitation method. Nanoparticles were functionalized with PEG. Then, a layer of CS polymer was anchored onto Fe_3_O_4_ nanoparticles by surface imprinting in the presence of propanamide as dummy template. While in the last work [49], Bagheri et al. prepared a dual magnetic hydrophilic MIP used as sorbent for simultaneous pre-concentration and determination of valsartan and losartan from urine samples. Fe_3_O_4_–COOH nanoparticles were prepared with a co-precipitation method and then a layer of CS was crosslinked as MIP layer onto Fe_3_O_4_–COOH nanoparticles in the presence of valsartan and losartan target molecules (Figure 2).

Barati et al. [50] developed a selective sorbent for separation and preconcentration of fluoxetine in pharmaceutical formulation, human urine, and environmental water samples, by using a supporting material made of magnetic CS and graphene oxide through a coprecipitation process (Figure 3). CS was first mixed with iron magnetic nanoparticles and subsequently glutaraldehyde and graphene oxide were added to the mixture. After, the graphene oxide/magnetic chitosan nanocomposite was collected using an external magnet. MIP was synthesized on the surface of the supporting material by coprecipitation polymerization of methyl methacrylate and ethylene glycol dimethacrylate (EGDMA) in the presence of fluoxetine. The multi-imprinting sites and the large surface area of the magnetic chitosan/graphene oxide allowed a fast sorption and desorption kinetics and high sorption capacity, while the magnetic property of the supporting material allowed simple, rapid, and efficient separation of sorbent.

In another example of core–shell systems, Surya et al. developed chitosan gold nanoparticles, acting as core, decorated MIP (shell) which was used to modify the glassy carbon electrode (GCE) for the preparation of an electrochemical biomimetic sensor to detect the ciprofloxacin antibiotic [51]. The simultaneous presence of Au nanoparticles and CS-MIP allowed to enhance sensitivity and selectivity respectively, for the detection of the antibiotic. First, gold nanoparticles were prepared in the CS solution from reduction of gold chloride. Then, ciprofloxacin-imprinted polymer was prepared from polymerization of methacrylic acid (MAA), in the presence of the template after mixing with a dispersion of chitosan-based AuNPs. Finally, to prepare the sensor, the chitosan gold nanoparticles, decorated with MIP, were dropped cast onto the GCE surface and dried.

Recently, Wang et al. prepared green IIPs membranes in aqueous phase via the synergy of three eco-friendly and low-cost functional monomers: gelatin (G), 8-hydroxyquinoline (HQ), and CS. They were applied as effective and recyclable adsorbents to remove Cu(II) from aqueous solution (Figure 4). Various important factors able to affect adsorption capacity including pH, temperature, and contact time, were investigated. Moreover, the fabrication of similar IIPs by using the same synthesis procedure but with other templates (Cd(II), Hg(II), Pb(II)) allowed to demonstrate that the utilization of three functional monomers could be developed into a general imprinting strategy towards various heavy metal ions through multi-point interactions [52].

CS-based ion imprinted microspheres with high specific surface area were also prepared in microfluidic apparatus [53,54]. He at al. synthesized Ca(II)–CS microspheres for heavy metal ions removal from wastewater. The optimization of the preparation process allowed them to obtain uniform microspheres in size, morphology, and perhaps nature. The dispersed phase feed was an acidified aqueous solution containing chitosan and Ca(II). The continuous phase feed was an n-octane solution with Span 80. The crosslinking bath was made up with n-octanol and 50 wt% glutaraldehyde aqueous solution and after phase separation of the mixture, the upper phase was out, and Span 80 was added as the crosslinking bath. The continuous phase and the dispersed phase were delivered into the microfluidic device, assembled with two thick plates and one thin plate in polytetrafluoroethylene, where the dispersed phase was introduced as small droplets. Crosslinking reaction with glutaraldehyde was conducted within the droplets collected in the crosslinking bath, as if the crosslinking agent entered the droplets by chemical potential driven interphase mass transfer (Figure 5). 

### 2.2. Cellulose

A biomass waste widely used in MIP preparation is cellulose, a very abundant polysaccharide in nature, synthesized by green plants, algae, and some bacteria species (*Acetobacter xylinum*, *A. hansenii*, *A. pasteurianus*, etc.). Cellulose consists of a linear chain of d-glucose units connected through ß-1,4-glycosidic bonds and characterized from the presence of inter- and intramolecular hydrogen bonds that confer high stability to the structure. Crystalline and amorphous regions characterize its composition. However, different treatments [55,56,57] can remove the amorphous components, obtaining cellulose in crystalline form. Bacterial cellulose is different in composition from plant cellulose and possesses high purity and major crystallinity. The presence of methoxy and hydroxyl groups in cellulose monomers facilitates their functionalization and confers to the natural polymer high adsorption ability but also high hydrophilic characteristics [58].

Biodegradable and biocompatible characteristics but also the high resistance to acids, basics and common solvents and the high stability in water, make cellulose usable in a lot of application fields [59]. For many years, cellulose fibers were used for paper production, as an energy source and in textiles. Recently, cellulose has been employed in pharmaceutical application [60,61], drug delivery [62,63], as sorbent material in solid phase extraction and in MIP technology [58,64,65,66,67]. The growing interest of the scientific community towards the preparation of new advanced materials by green routes, prompted researchers to use cellulose as backbone-based materials to prepare green and biodegradable MIPs. In the last years, several papers on the usage of cellulose and cellulose derivatives as key element in the preparation of biodegradable and biocompatible MIP and IIP were published [56,59,68,69,70,71,72,73,74,75,76,77,78,79,80,81,82,83]. The prepared cellulose-based imprinted polymers were used in many application fields ranging from drug delivery, separation science, and also in environmental analysis of pollutants. A common cellulose modification adopted in many works is the addition of carboxylic residues on the cellulose backbone.

An interesting work conducted by Farenghi and co-workers, consists in the preparation of a water compatible cellulose-based MIP for Furosemide slow release in drug delivery studies [70]. First, microcrystalline cellulose was modified in sodium carboxymethyl cellulose (CMC) and used for the CMC-MIP design. In Figure 6, a synthetic route for CMC-MIP preparation was reported. After modification of microcrystalline cellulose, CMC was added in a preassembly solution containing the template (Furosemide) and the functional monomer acrylamide (AA). Then, CMC was crosslinked using 2-hydroxyethyl methacrylate as a co-monomer and ammonium peroxydisulfate as the CMC activating agent. 

In order to evaluate the binding performance of the polymer synthesized, binding studies were conducted revealing higher efficiency and selectivity of CMC-MIP compared with the corresponding non imprinted polymer (NIP). Finally, in vitro release studies, conducted in a phosphate buffer solution at 37 °C in order to simulate arterial blood conditions, showed a higher loading capacity of CMC-MIP at the equilibrium state with a good control of the drug release rate compared to the corresponding CMC-NIP [70].

Carboxymethyl cellulose was also used as starting material for the preparation of a thiol-imprinted polymer for selective adsorption of Hg(II) [77]. In this case, CMC was crosslinked and stabilized with epichlorohydrin through an amide reaction. Then, carboxylic moieties of crosslinked CMC were modified by the thiol ligand Cysteamine in order to favor the complexation with Hg(II) ions. After the addition of the template ion, the functional monomer methylene bis acrylamide and ammonium peroxydisulfate as initiator, the reaction was conducted at 70 °C for 12 h. The adsorption ability of the polymer was tested, and a maximum adsorption capacity of 80 mg g^−1^ was obtained. Moreover, polymer efficiency was evaluated in real samples showing recoveries for Hg(II) of 86.78%, 91.88%, and 99.10% in wastewater, ground water, and tap water, respectively.

Carboxylated cellulose nanocrystals, coupled with magnetic materials and combined with surface-imprinted polymer technique, were studied from Hu and co-workers to obtain a new adsorbent with high selective adsorption of fluoroquinolones in water [56]. 

Recently, Lin et al. [73] prepared for the first time a photo-responsive cellulose-based imprinted polymer (PR-Cell-MIP) as intelligent material for selective adsorption of typical pesticide residue, 2,4-dichlorophenoxyacetic acid. In this work, cellulose was first esterified to obtain a macromolecular initiator (Cell-ClAc). In order to confer photo-responsive characteristics, an Azo-containing functional monomer with carbon double band (Azo-CB) was synthesized and combined with the template in a pre-polymerization step. After, the photo-responsive cellulose-based imprinted adsorbent was synthesized via surface-initiated atom transfer radical polymerization (Figure 7). The results obtained showed remarkable adsorption performance (Q_max_ = 11.601 mg L^−1^ with 89.03% of equilibrium template bindings), selectivity, stability, and reusability (8 cycles) of PR-Cell-MIP compared with the corresponding non imprinted polymer. Moreover, the presence of the Azo functional monomer, favors a green regeneration of the polymer through UV light.

Recently, many works have published in order to obtain efficient separation systems for heavy metals and rare earth element extraction. For this purpose, cellulose and their derivatives were widely used [71,77,78,82,83,84,85].

An innovative methodology to prepare a new generation of IIPs for heavy metal ions extraction was introduced for the first time from Fattahi and co-workers [71]. Biological nanocrystal cellulose, obtained through acidic treatment of cotton wool, and cetyltrimethylammonium bromide were used as hard and soft templates respectively, in order to fabricate a new silica-based imprinted mesoporous polymers, via dual template method, for the simultaneous extraction of cadmium and lead ions from river water and fish muscles by micro solid phase extraction.

In a recent work, a stable and insoluble material was obtained using rice straw as support for cellulose-based lanthanum IIP preparation combining the imprinting technology with activators produced by electron transfer for atom transfer radical polymerization (AGET-ATRP) [85]. Hydroxyl groups present on the rice straw reacted with 2-bromoisobutyryl bromide in presence of triethylamine in order to introduce the alkyl bromide initiator for surface-initiates AGET-ATRP. After, the polymerization was completed using the AGET-ATRP technique, with Cu(II) *N*,*N*,*N*’,*N*’,*N*”-pentamethyldiethylenetriamine (Cu(II)/PMDETA) catalyst that was reduced in Cu(I)/PMDETA in the presence of ascorbic acid. The addition of functional monomer *N*,*N*’-dimethylaminoethyl methacrylate (DMAEMA), crosslinking agent EGDMA, and template ion (La^3+^) completes the preparation of the straw-supported imprinted polymer that was subjected to washing steps in order to remove the template ions from cavities present in the polymeric structure. The density of IIP polymer cross-linkage on the rice straw surface exerted great influence on the sorption property with a maximum sorption capacity of 125 mg g^−1^ obtained in several minutes.

In another work, Wu and co-worker prepared for the first time a three-dimensional macroporous wood-based selective separation membrane decorated with well-designed Nd(III)-imprinted domains in order to recover Nd ions [82]. First, a polydopamine (PDA)-modified layer was synthesized on the basswood surface in order to obtain PDA-based basswood membrane (PDA@basswood) that was further modified through the addition of KH-570 to achieve polymerizable double bonds for the following two steps (Figure 8). Through a two-step temperature free radical polymerization process, obtained using Nd(II) ions as template, two functional monomers (MAA and AA) and EGDMA as crosslinking agent, a 3D-based ionic imprinted membranes (3DW-IIMs) system was obtained. Excellent results in terms of rebinding capacity (120.87 mg g^−1^); adsorption kinetics and permselectivity coefficients (more than 10) were obtained. Moreover, it was found the capability of the PDA-modified layer to enhance the rebinding performance of the 3DW-IIMs.

### 2.3. Activated Carbon

Activated carbon is essentially a carbonaceous material with specific physicochemical properties, which make it attractive for different applications [12,86]. It is a black solid substance typically amorphous, microcrystalline, and non-graphite which cannot be identified with any determined chemical formula. Depending on its physical form, activated carbon can be characterized as powder, granule, extract, pellet, fiber, cloth, and so on. Its chemical structure has heteroatoms like oxygen, hydrogen, sulfur, and halogen as atoms or as functional groups chemically bonded into the structure. Among them, oxygen is the most common one and it is mainly included in carboxyl, phenol, carbonyl, and lactone functional groups into the inner or external surface of the material. The quality, the properties, and the characteristics of the chemical structure and surface chemical groups of the activated carbon depend on the precursor source and the oxidative activation treatment. Common physicochemical properties of activated carbon are high porosity, surface area and degree of surface reactivity together with high mechanical strength and physicochemical stability. Thanks to these properties, activated carbon has been extensively used for different applications where for example high adsorption capacity, hardness, or abrasion resistance are needed.

Precursor source or starting material of activated carbon can be either natural or synthetic carbonaceous solid one. The most common sources on the market are petroleum residues, wood, coal, peat, and lignite. Most of them are expensive and not renewable. Recently, other precursors have been taken into consideration, and many research works have been focusing on the activated carbon production starting from agricultural waste and lignocellulose materials. Even if carbon content in these previously mentioned precursors is lower than the carbon content in coal, anthracite, or peat, biomass waste is very attractive because of its relatively low cost, its green aspects (as the waste could be converted into capital), and it can be considered renewable and capable to solve environmental concern. Several biomass sources have been used to produce activated carbon such as coconut shell, sugarcane bagasse, coffee ground, hazelnut shell, olive waste cake, rice husk, groundnut shell, hazelnut bagasse, kenaf fiber, or coconut husk [86,87,88].

Activated carbon preparation from biomass waste can be realized through a physical or a chemical treatment. Physical treatment is carried out in two steps: firstly, the carbonization occurs by a thermal process, heating in a range of 400 to 850 °C and successively the activation process is made by using steam or carbon dioxide at a temperature of 600–900 °C. By contrast, in chemical treatment only a step is needed: an activating agent is added to the biomass waste in order to allow starting material dissolution and crosslinking formation, and a thermal process under inert atmosphere is made.

Activated carbon has been applied in several fields as supported from the huge literature published in the last decades such as water treatments, chemical and pharmaceutical industries, catalysis, energy storage, batteries, nuclear power stations, and supercapacitors [87,88,89,90]. 

Recently, some papers focusing on the use of activated carbon in molecular imprinting technology have been also published [91,92,93,94]. In fact, advantages of typical activated carbon properties especially high porosity, high surface area, and the presence of functional groups (carboxyl, carbonyl, or phenol) on its surface as well as environmentally friendly are combined with selectivity typical of molecular ion imprinting technology.

It is well known that activated carbon are excellent adsorbents for many heavy metal ions and organic pollutants even if they are often unable to distinguish between different metal ions or organic pollutants. In this contest, Zhang et al. in a recent work improved the selective adsorption capacity of activated carbon, called biochar, by using ion imprinting technology [91]. The authors used biochar, prepared by processing agricultural waste under slow pyrolysis at 600 °C for 10 h, as the substrate material to obtain an ion-imprinted functionalized sorbent for Cd(II) from wastewater. Biochar was first activated with hydrochloric acid and then it was modified using 3-mercaptopropyltrimethoxysilane as the surface conditioning agent and epoxy-chloropropane as the crosslinking agent, in the presence of a hydrate chloride salt of cadmium(II) in ethanol in order to prepare the ion-imprinted sorbent (IB). The obtained results demonstrated the higher selectivity for Cd(II) of the imprinted biochar (IB) even in the presence of Co(II), Pb(II), Zn(II), and Cu(II) interferences.

Glycoproteins are crucial in many biological events, such as protein folding and nerve conduction, and some glycoproteins can be used as disease biomarkers. However, selective detection of glycoproteins at trace level in complex biological matrices remains an urgent issue. To address this, in a very recent work, Ding et al. developed a very interesting biomass activated carbon-derived imprinted polymer (BAC@PEI/PBA/MIPs) for selective recognition of glycoprotein [94]. They combined biomass active carbon with surface imprinting technique to obtain a biowaste-derived imprinted polymer for specific albumin recognition through an easy synthetic strategy starting from low-cost materials. The BAC@PEI/PBA/MIPs was synthesized using waste tea derived carbon as matrix, albumin chicken egg as template, and dopamine as functional monomer. In order to prepare activated carbon, dried waste tea was mixed with potassium ferrate and annealed at 800 °C for 1 h under nitrogen. Then, carboxylation was carried out with sodium nitrate and potassium permanganate in acidic environment. Moreover, the binding capacity was significantly increased thanks to the branched polyethyleneimine (PEI) which was covalently bonded on the activated carbon surface before to carry out the surface imprinting, in order to increase the number of boronic acid sites. After that, the boronic acid molecules were immobilized via amine aldehyde condensation reaction (Figure 9). As the molecular weight of PEI can affect the binding capacity, Ding et al. prepared different BAC@PEI/PBA/MIPs with PEI at various molecular weights and the binding capacity was evaluated. The binding performance of BAC@PEI/PBA/MIPs compared with some reported glycoprotein-MIPs was superior, with a faster mass transfer rate of the BAC@PEI/PBA/MIPs than the most reported MIPs, even if the imprinting factor need to be improved.

Taking into consideration the huge amount of carbon dioxide released in the atmosphere, it is vital to develop advanced carbon dioxide adsorption materials for protecting the environment. To this aim, carbon capture and storage represents a convenient environmentally friendly way for reducing atmospheric CO_2_. In this context, in another very recent work, Su et al. prepared a CO_2_-imprinted polymer on the surface of activated carbon for effective capture of CO_2_ from flue gas [92]. Sustainable carbon was derived from sunflower heads while the surface imprinting technique was used with ethanedioic acid and AA as template molecule and functional monomer, respectively. The authors optimized the imprinted polymer by changing the ratio of KOH to sunflower-based activated carbon, carbon dosage, and adsorption temperature. Adsorption, selectivity, and regeneration properties of the optimized imprinted polymer were evaluated. A maximum CO_2_ adsorption capacity of the adsorbent of 1.71 mmol g^–1^ was found. Selective adsorption performance in the presence of H_2_O as well as in simulated flue gas was demonstrated. Moreover, regeneration by N_2_ purge at 120 °C showed a CO_2_ adsorption capacity decreasing of 11% after five adsorption/desorption cycles.

Finally, it is interesting to describe the work of Saputra et al. [93]. The authors reviewed the synthesis of biosensors from teak lignocellulosic material. They describe a type of activated carbon preparation technique with intrinsic products such as the stable nature of chemical reaction, porous structural hierarchy, and decreased detection limits. Nanocarbon was obtained into a drum-kiln at 400–500 °C for 7–8 h. Then, the activated carbon was produced at 800 °C for 1 h with H_2_O and potassium hydroxide activator. MIP and NIP synthesis techniques were applied for biosensor production for melamine detection, while potential measurements and detection limits were used to measure product performance. The results showed an optimal formula for a mixture of 15% MIP, 45% carbon, and 40% paraffin which produced a Nernst factor of 49.7 mV decade^−1^ and detection limit of 1.02 × 10^−6^ M.

### 2.4. Carbon Dots

Fluorescent carbon dots (CDs) are emerging quantum nanomaterials derived from carbon materials that have attracted significant attention compared to conventional semiconductor quantum dots thanks to more interesting properties such as bright good stability, fluorescence, noticeable electrochemical activity, easy synthesis, proper surface functionalization, and especially non-toxic features [13,95]. 

They appeared for the first time in 2004 in a work of Xu et al. [96], and they were named as carbon dots two years later by Sun et al. [97]. From their discovery until now, the scientific community has been devoting enormous consideration to CDs due to their excellent characteristics. They are well-dispersed fluorescent carbon nanospheres with particle size below 10 nm. Several great properties are typical of CDs such as good biocompatibility, easy chemical surface modification, high chemical inertness, noticeable electrochemical activity, green and easy synthetic strategy, and good water solubility. Special attention should be given to their fluorescent characteristics which are also possessed by traditional semiconductor quantum dots, which make them good competitors: a broad absorption spectrum, multiple generations, high photostability, and fluorescent intensity, and their emission spectrum generally changes by changing excitation wavelengths or by changing nature of functional groups on the surface. Similar to quantum dots, the emission of carbon dots is related to their size. Thus, valuable CDs have been used as fluorescent probes for biological or environmental detection [95]. When CDs are included in a sensor system, the main challenge is to find the proper matrix which can preserve the photoluminescence activity and avoid leaking of carbon dots. CDs are also finding application in other fields such as cell imaging, in vivo imaging, drug delivery [98], photocatalysis [99], multicolor light-emitting diode production [100], and energy conversion and storage [101].

CDs properties such as morphology, fluorescence, and luminescence efficiency depend on the preparation method and on the nature of starting material. Development of synthetic strategies to get CDs with defined properties is still in progress since it is only at an early stage. There are two main synthetic pathways followed to produce CDs: top-down and bottom-up approach. The first one goes from larger carbon materials (activated carbon, graphite, carbon nanotube, large-size graphene, etc.) to carbon dots by using electrochemical oxidation, arc discharge or laser ablation technique. Some disadvantages of this method such as the use of strong experimental conditions, time consuming operation steps and high-cost equipment have limited its use. On the other hand, the bottom-up method converts small molecules into carbon dots through carbonization and passivation steps by using microwave-assisted, pyrolysis or solvothermal, ultrasonic-assisted methods. Some advantages typical of this method such as cost-effectiveness, easy operation and simple equipment requirements allowed an extensive employment of it.

Starting materials for CDs preparation are very widespread and even if organic or inorganic carbon sources are both available, due to some limits of the inorganic one, organic carbon sources represent the main font of carbon dots. Between the different organic carbon classes considered, biomass waste has been recently used as raw material for CDs synthesis for the reasons and advantages widely discussed in the current review. Numerous biomass wastes have been used for carbon dots preparation such as sugarcane bagasse char, waste food, orange or lemon peels, onion waste, coffee grounds, tea or rice residues, wheat bran or straw, bamboo leaves, coconut husks, lignite, and so on [13,102]. 

CDs are good candidates to be combined with molecular imprinting technology as specific recognition site, selectivity, and stability typical of MIPs enable them to enhance the unique optical and electronic properties of CDs. In fact, one of the main limits of CDs in sensor application is their lack of selectivity to the analyte as long as the presence of interferences in complex matrices responsible for quenching fluorescence emission spectrum of CDs. Thus, with the aim to enhance selectivity, sensitivity, and anti-interference ability, some attempts to apply CDs in the MIP field have been made in the last few years. Note that the relevance and success of the carbon dot-based MIP research area has been demonstrated from the publication of some reviews in 2020 with this specific topic where a description in deep several studies of CDs combined with MIP can be found [20,103,104], where optical/ fluorescence sensors represent the main explored application of CDs combined with MIP. In this context, even if Ansari et al. [20] dedicated a paragraph for CDs derived from biomass waste, they reported only two examples where CDs were synthesized from sweet potato peels and cedrus, respectively [105,106]. Thus, herein other relevant studies recently appeared in literature will be described highlighting some aspects of carbon dots synthesis from biomass waste. As it can be seen, CDs can be utilized as a support material for surface imprinted polymers to develop selective and sensitive sensors. 

It can be observed that Liu adopted a green approach for CDs preparation starting from sweet potato peels [105] and longan peels [107] as carbon biomass waste source by using a hydrothermal synthesis, avoiding toxic solvents or complicated procedures. While in their previous work, Liu et al. developed a MIP-coated CDs by using a hydrothermal synthesis in water heating at 200 °C for 3 h in autoclave, in a successive paper published in 2020, Liu et al. used an easier method to obtain CDs from biomass waste, mixing longan peels and water and heating at 200 °C inside a high-pressure microwave, which allowed to decrease preparation time at 30 min. The carbon dots coupled with restricted access materials and molecularly imprinted polymers (CDs@RAM-MIPs) composites were prepared by using CDs, metronidazole, AA and EGDMA as fluorescent materials, template molecule, functional monomer, and crosslinker respectively. Moreover, glycidyl methacrylate was also used as co-polymerization functional monomer to produce a large amount of hydroxyl groups and to make the composites more hydrophilic. The results demonstrated that CDs@RAM-MIPs can be used as specific and selective probes for trace detection of metronidazole in serum samples.

Sun et al. fabricated a fluorescent probe based on MIPs combined with carbon quantum dots used for selective adsorption of mesotrione with fluorescence quenching to determine the analyte in corn. Carbon quantum dots were prepared through a hydrothermal method using mango peels as carbon source, and the whole synthesis procedure was green without chemical reagents. Then, CDs were encapsulated into MIPs by using sol-gel technology (Figure 10). The MIP probe was successfully applied to determine mesotrione in a complex matrix as a low detection limit of mesotrione of 4.7 ppb and a high selectivity with an imprinting factor of 5.6 were found [108].

In a recent paper, Kazemifard et al. described the use of rosemary leaves, as a carbon source, to synthesize CDs [109]. Following the rules of green chemistry related to the elimination or minimization of the use or production of harmful chemicals, the authors used only water and rosemary leaves. The powder of dried leaves was added to water and heated at 180 °C for 12 h. Carbon dots were purified by centrifugation and filtration of the supernatant. Then, the surface of the CDs was chemically modified to obtain a crosslinker for MIP in order to prepare an optical probe for the determination of thiabendazole (TBZ) and it was successfully applied for the quantification of TBZ in apple, orange, and tomato juices. In detail, a silica shell using tetraethoxysilane (TEOS) was stabilized on the surface of CDs via reverse microemulsion technique. Following, MIPs were synthesized in the presence of TBZ as a template, using 3-aminopropyl triethoxysilane and TEOS as a functional monomer and a crosslinker, respectively.

In a work of Demir et al. [110], for the first time a core–shell CD-MIPs as a biocompatible optical imaging tool for targeted bioimaging of cancer cells, was described. Glucuronic acid was used as template by using the epitope approach for hyaluronan recognition, a biomarker for specific cancers. *N*-doped CDs were prepared by hydrothermal synthesis using starch as carbon source and l-tryptophan as nitrogen atom in water. The mixture was heated at 160 °C for 12 h and then centrifuged. The supernatant was filtered, and CDs were purified by using size-exclusion chromatography. Then, a thin shell of MIP was photopolymerized around the CDs surface by using CDs light emission as light source for photopolymerization. In this work, the biotargeting and bioimaging of hyaluronan on defined human cervical cancer cells, using CD core–MIP shell nanocomposites, was demonstrated. A brief overall description of the study is shown in Figure 11.

Finally, it is interesting to describe a work that shows the great potential of MIPs combined with carbon dots in the photocatalysis field. In most reports, CDs/semiconductor hybrid photocatalysts exhibit excellent optical properties and photogenerated carriers transfer characteristics [99]. Taking into account this CDs property, Yu et al. in 2021 provided a molecularly imprinted black MIP-TiO_2−x_/CQDs nanocomposite material to efficiently photodegrade target pollutants in wastewater. CDs were obtained from lignite dispersed in H_2_O_2_ and formic acid, stirred at 80 °C for 3 h. After a filtration, diethylenetriamine was added and heated at 180 °C for 6 h. The composites were prepared via a facile two-step hydrothermal calcination method by using methylene blue as template. The unique optoelectronic properties of carbon dots can be more effectively utilized by introducing it into black MIP-TiO_2−x_. The insertion of carbon dots into black MIP-TiO_2−x_ not only improved the optical absorption ability, but also promoted the formation of Ti^3+^ ions and oxygen vacancies in MIP-TiO_2−x_ nanocomposites and effectively enhanced the separation of photogenerated (Figure 12). In fact, the results showed an improvement of photocatalytic activity of MIP-TiO_2−x_/CDs to methylene blue under visible light [26].

### 2.5. Cyclodextrins

Starch represents a low-cost biomass resource that can be easily converted into valuable materials through eco-friendly processes with low environmental impact. Enzymatic conversion of starch sources like potato, corn, rice, tapioca, and wheat [111,112,113] determines the formation of cyclodextrins that represent natural polymers rich in active functional groups that can establish good affinities with numerous organic and inorganic compounds. Cyclodextrins are a family of cyclic oligosaccharides composed from β-(1,4)-linked glucopyranose subunits. Depending on the number of sugar units involved, cyclodextrins can be classified in α-, β-, and γ-cyclodextrins (α-, β-, γ-CDs) that contain respectively six, seven and eight glucopyranose subunits [114]. The diameter of cyclodextrins enhances with the number of monomers involved in their structure, assuming a characteristic form of a truncated cone with the presence of hydrophilic groups oriented outside the cone and hydrophobic groups inside. Thanks to their unique properties to form inclusions of specific guest molecules, in the last decade they were used in different application fields such as enantiomer separation and drug delivery [115]. Moreover, all functional groups present in cyclodextrins can be easily modified in order to introduce more desired functional groups to modulate selectivity, solubility, etc. These characteristics make cyclodextrins suitable for their use as functional monomers in MIP technology. In particular, β-CDs are the most applied in MIP because their poor solubility in water facilitates its low-cost recovery through crystallization, compared with α- and γ-CDs that require expensive and time-consuming chromatographic technologies. Moreover, its chemical structure presents more discrimination ability, and thus eligiblity for MIP preparation [115]. Different works showed that the presence of cyclodextrins enhances the efficiency and the selectivity of the MIP towards the template molecule [116,117]. Finally, the hydrophilic groups present outside of cyclodextrins, improve the application of cyclodextrins-based MIPs in aqueous media [118], overcoming one of the main limitations of MIPs.

In the last decade, some reviews were published on the use of cyclodextrins in MIP preparation, pointing out the interest of the scientific community about this topic [115,118,119]. Very recently, Zhao et al. focused their work on the advances on cyclodextrins-based MIP applications ranging from their use in separation science, sensing and drug delivery [118]. 

Cyclodextrins can be used directly to act as specific functional monomers in MIP preparation [120,121,122] or can be modified in order to enhance the interactions with the template molecule. Moreover, they can also be used as binary functional monomers in combination with other monomers [123,124,125,126,127,128,129]. A common modification used consists in the addition of acryloyl and allyl moieties [115,130,131,132,133]. A complete description of different synthetic routes to modify cyclodextrins was widely discussed from Zhao [118]. For this reason, we will describe only some recent works highlighting the possible use of cyclodextrins as functional monomers and supporting materials in MIPs with interesting applications.

An application field widely studied regards the research of innovative materials for environmental remediation of pollutants. To this aim, a recent work conducted by Mamman et al. concerned the use of a modified β-CD as functional monomer, to prepare a magnetic MIP (MMIP) for bisphenol A (BPA) retention in aqueous media [117]. First, MAA-β-CD was synthesized by a classical synthetic route mixing MAA, β-CD and toluene 2,4-diisocyanate (0.5/0.5/1; *v*/*v*/*v*) in dimethylacetamide (DMAC). Then, the magnetic polymer was prepared using MAA-β-CD as functional monomer, BPA as template, trimethylolpropane trimethacrylate (TRIM) as crosslinking agent, and benzoyl peroxide (BPO) as initiator. In order to confer magnetic properties, the particles obtained were mixed with iron chloride in water. A magnetic NIP (MNIP MAA-βCD) was also prepared using the same synthetic route but without the presence of the template molecule. Moreover, in order to evaluate the influence of β-CDs in the imprinting capacity, a control polymer was prepared using only MAA as functional monomer. Adsorption performances were conducted in PBA water solution. Different parameters that influence the adsorption capacity of polymers prepared were analyzed. In particular, pH, concentration of polymeric particles, contact time, and temperature were studied. The results obtained, comparing the adsorption performance of all polymers prepared, showed an optimum adsorption time of 60 min at pH 8 with an analyte concentration of 10 mg L^−1^ and an adsorbent dose of 20 mg for MNIP MAA-βCD. It is interesting to note how the presence of β-CDs influences the adsorption performance enhancing their capability to trap BPA. These results can be due to the possibility of BPA to form an inclusion complex with β-CD during the recognition process enhancing the adsorption behavior. Indeed, compared to MMIP MAA, MMIP MAA–βCD showed higher initial sorption rates (7.4906 mg g^−1^ min) obtained from a pseudo-second-order kinetics model with low values of time required for the adsorption. This can be attributable to the presence of β-CD moiety in the internal surface of MMIP MAA–βCD which determines the presence of pore with large size and low volume of polymeric matrix improving the mass transfer of BPA [117].

An innovative approach to prepare functionalized cyclodextrins consists in the possibility to use luminescent materials to modify their hydroxyl groups in order to form a probe capable to change its optical signal in the presence of microenvironmental changes due to the bond with a target compound. An interesting work made by Shi and co-workers [134] describes the preparation of surface-imprinted β-cyclodextrin-functionalized carbon nitride nanosheets (CNNS) for fluorometric determination of sterigmatomycin. CNNS were obtained from the exfoliation of graphitic carbon nitride (g-C_3_N_4_) and possess interesting fluorescence properties. In a first step, after preparation of CNNS, β-CD were modified by liquid phase exfoliation in order to obtain β-CD/CNNS [135] that was successively used as support for sterigmatomycin-MIP preparation (Figure 13).

β-CD/CNNS were first dissolved in dimethylformamide and stirred with the template at 60 °C for 3 h. After, functional monomer (MAA), crosslinking agent (EGDMA), and the initiator 2-2’-azobisisobutyronitrile (AIBN) were added and stirred to obtain an innovative MIP@β-CD/CNNS for fluorescent detection of sterigmatomycin. Adsorption studies demonstrated a larger adsorption capacity of MIP@β-CD/CNNS (86 mg·g^−1^) compared with the corresponding NIP and with the different components of polymer (β-CD, g-C_3_N_4_ and β-CD/CNNS). Moreover, MIP@β-CD/CNNS shower less time (25 min) to reach an equilibrium in the adsorption compared to the corresponding NIP. Also in this case, this is due to the presence of β-CDs in the polymeric matrix that determine a fast mass transfer rate of the surface imprinted polymer. Then, fluorescence resonance energy transfer (FRET) technology was used to verify the real use of MIP@β-CD/CNNS as probe for detection of STG.

Results obtained revealed excellent photoelectric properties of the MIP prepared that can be used as optical probe but also as an efficient adsorbent and catalyst material for more applications.

As punctually described from Zhao, cyclodextrin-based MIPs were widely used to prepare electrochemical sensors [118]. Often ß-CDs are used as functional monomer, can be electropolymerized on electrode surfaces without the addition of initiators and crosslinking agents and it can act as supporting material in MIP preparation.

A recent work describes a molecularly imprinted polymer sensor for horseradish peroxidase (HRP) prepared through an electropolymerization of ß-CDs on the surface of a glassy carbon electrode [136]. In this study, Poly beta-cyclodextrins P(ß-CDs) were used as supporting material and the reaction was carried out combining electropolymerization of ß-CDs and the molecular imprinted technology. Electropolymerization represents a phenomenon that requires the oxidation of a monomer to form a radical cation applying an oxidation potential. In this context, β-CDs (Figure 14).

The results obtained showed excellent electro-conductivity of β-CDs that facilitated electron transfer of P(ß-CD). The imprinted polymer sensor showed good detection performance of HRP with a linear range between 0.1 mg mL^−1^ to 10 ng mL^1^ and LOD of 2.23 ng mL^−1^. Moreover, the imprinted biosensor, applied to real samples, showed a good electro-catalytic activity towards the reduction of H_2_O_2_ that was evaluated in a concentration range of 1 to 15 μM with a detection limit of 0.4 μM using chronoamperometry technique. For these reasons, the developed bio imprinted sensors can be used as platform for biomedical analysis of hydrogen peroxide concentration in biological matrices such as human plasma.

### 2.6. Biomass Waste Extracts in MIPs Preparation

Other important natural starting materials obtained from agri-food processing are waste extracts. There are several works and reviews that have focused their attention on the possible use of waste extracts as green chemicals to reduce environmental pollution [137,138,139,140,141,142]. Their compositions are characterized by numerous important biomolecules such as sugars and polyphenols that can act as reducing agents for metal salts, but also as natural stabilizers. The action of these two classes of biomolecules determines the formation of stable metal nanoparticles obtained through an eco-friendly process which reflects the principles of green chemistry. In the last years, there are a lot of works that focused their research on the use of waste extracts as a driving force for the design of greener and safer protocols for metal nanoparticles preparation [138,142,143,144,145,146]. Recently, the advantages of metal nanoparticles were combined with the specificity of MIPs in order to confer higher sensitivity, magnetic, and also optical properties due to the plasmon resonance typical of metal nanoparticles such as gold and silver. 

A common synthetic route to prepare metal nanoparticles requires the use of a chemical reducing agents such as borohydrides, sodium borohydride that are harmful for the environment and limited the use of nanoparticles obtained in medical application because of their toxicity or safer sodium citrate and alcohols; but, in any case, chemical substances need to be added to prevent aggregation processes. Recently, some works combining the advantages of metal nanomaterials with the selectivity of MIPs were published [146,147,148,149,150]. Only one of these was made using metal green nanoparticles obtained from waste extracts in MIPs preparation [150]. 

In their work, López et al. prepared, for the first time, green iron nanoparticles using eucalyptus extract (GNP) to confer magnetic properties at a series of MIPs selective for seven non-steroid anti-inflammatory drugs (diclofenac sodium, indomethacin, acetaminophen, naproxen, ketoprofen, ibuprofen, and mefenamic acid) and their adsorption performances were compared with the same imprinted polymers prepared using iron nanoparticles synthesized by chemical route (CNP) [150,151,152]. GNP were prepared in a bath at 70 °C, adding the eucalyptus extract to an iron solution. After, GNP was precipitated by the addition of NH_3_, neutralized with ultrapure water and dried under vacuum. Then magnetic MIP-GNP and magnetic MIP-CNP were prepared by mixing the template and iron nanoparticles (GNP or CNP) in a porogenic solvent. After the addition of the functional monomer, the crosslinking agent (EGDMA) and the initiator (AIBN) the reaction was conducted at 60 °C. Comparing the adsorption performances of all MIPs prepared it is obvious the influence exerted from GNP in the binding of the template. Although magnetic MIP-CNP reaches equilibrium in shorter times than magnetic MIP-GNP, the presence of organic matter in the extract solution acts as capping agent enhancing the adsorption performance of MIP-GNP. Indeed, the adsorption capacity with green nanoparticles was increased by more than 100% in all cases, in relation to the adsorption value of the chemical nanoparticles. 

In another work lemon juice was used as natural reducing agent instead of chemical compounds such as sodium borohydride to prepare vinyl modified reduced graphene oxide-based magnetic and bimetallic (Fe/Ag) nanodendrite (RGO@BMNDs) as innovative platform for MIP technology [151]. Citrus plants contain high amounts of antioxidant compounds and can act as low cost, easily available, and natural reducing source. Firstly, Graphene Oxide (GO) was prepared from graphene powder through an oxidation process by using strong oxidizing agents. In the final step graphene sheets were obtained and used as support for the synthesis of bimetallic (Fe/Ag) nanodendrites through a facile Fe nanoseed-induced method. At first, zero valent iron was formed on the RGO surface by using lemon juice as natural reducing agent, followed from a displacement reaction that partially replace the Fe with Ag, in order to obtain a bimetallic nanodendrite structure (Figure 15). These steps were conducted without the use of harmful compounds or harsh physical conditions respecting the principle of green chemistry. Then, synthesized RGO@BMNDs with high catalytic activity and adsorption capacity, were used as platform for the preparation of an imprinted polymer modified electrochemical sensor for detection in human blood serum and in plasma samples of ultra-trace of the analyte pyrazinamide (PZA), a drug for the control of tuberculosis disease, with a limit of detection equal to 6.65 pg L^−1^.

A facile and low-cost photosynthetic route was used from Essawy to prepare silver imprinted zinc nanoparticles (AGZnONPs) by using aqueous extract of guava leaves for nanoparticles preparation instead of undesirable chemical reagents. In particular, AgNO_3_ and ZnO aqueous solution were added dropwise to guava leaves extract and heated at 70 °C under vigorous stirring. Nanoparticle formation is confirmed from the color change to brownish and their photocatalytic activity was demonstrated estimating the degradation rate of methylene blue dye [152].

Other biomass resources obtained from the agri-food industry were used in some works as starting material to prepare new biobased MIPs [153,154].

In a recent work epoxidized soybean oil (ESOA) was used as an innovative crosslinking agent in the preparation of a biobased MIPs employed as a biopesticide delivery system [153]. The crosslinking agent represents the major component of MIP and for this reason its substitution with natural compounds represent a great challenge. Due to their composition, vegetable oils are considered today renewable natural materials useful in MIPs preparation and can confer stability to the polymers. In this work, resveratrol, with important antifungal activity against several phytopathogens, was used as template and mixed with functional monomer (4-vinylpyridine or 1-vinylpyridine), crosslinking agent (EOSA or EGDMA), and initiator (AIBN) in chloroform. The reaction was conducted for 50 °C for 24 h. Resveratrol-MIP using EOSA was also prepared by emulsion polymerization and the results were compared. Results obtained showed a maximum adsorption capacity of EOSA-base MIP, prepared by bulk polymerization, equal to 34.2 μmol g^−1^ that was higher than the adsorption values obtained with the corresponding imprinted polymer prepared using EGDMA instead of EOSA. Moreover, it was demonstrated the ability of EOSA-based MIP to release resveratrol in aqueous medium, exhibiting high sensitivity to a fungal lipase, due to the enzymatic degradability of the polymer. This result is very interesting for future application in agricultural treatments because it suggests eco-friendly characteristics of the material that can be naturally degraded.

Starch extracted from *Solanum tuberosum* was used as copolymer with polyvinyl alcohol (PVA) for the preparation of an ion imprinted Starch/PVA polymers for Thorium (IV) adsorption [154]. Starch contains a large numbers of hydroxyl groups and was incorporated in an ionic liquid (1-butyl 3-methyl-imidazolium tetrafluoroborate ionic liquid), with PVA and glutaraldehyde (crosslinker) and the reaction was conducted at 45 °C. After addition of thorium, the solution was subjected to electrospinning at 11.5 kV voltage and 0.54 mL min^−1^ flow rate. Nanofibers were collected and treated with HCl (37%) solution at pH 5 to eliminate thorium ions and then washed with ultrapure water. Results obtained suggest a high sorption efficiency (87%) of the polymer at ecologically optimum conditions, at pH 7 and 0.5 g of adsorbent dose, with a best compliance with Langmuir model.

### 2.7. Summary

All the studies reported in this review on various waste sources used in MIPs preparation have been summarized in Table 1.

## 3. Conclusions

This review provides an extensive overview on the use of different kind of biomass waste derivatives as backbone materials in MIP preparation, pointing out the most recent works that developed greener analytical protocols of synthesis to reduce environmental pollution. A complete description of strategies to replace typical highly toxic chemicals generally required in MIPs, with natural polymers obtained from biomass wastes, such as chitosan, cellulose, or cyclodextrins, was made. In particular, it was underlined the possibility to modify these natural resources in order to obtain new green substrates for MIPs preparation that can be used as innovative functional monomers, initiator, crosslinking agent, or supporting materials. Activated carbon and carbon dots were also deepened for their important chemical properties, the high availability and low costs for production and used in association to MIPs in order to enhance their selective absorption or their electrochemical properties, respectively.

The works analyzed in this review show how the introduction of biomass wastes in MIPs preparation can determine an increase of adsorption and selectivity performance and also greater sensitivity and morphology control, compared to the same polymers prepared using common synthetic routes.

We hope that the results highlighted in this review will be useful to stimulate the use of more sustainable preparation strategies for MIP production in order to exploit eco-friendly resources, such as biomass waste, easily available in nature.

Although numerous examples of preparation of MIPs by using different biomass waste have been recently described in literature, many challenges still exist. First of all, the use of biomass waste in combination with other materials, such as nanoparticles or magnetic materials, represent a great potential since it allows to obtain novel composite materials with unique properties. However, it requires to apply improvement in the green concept to the whole design process. Second, it is still a big challenge to realize the large-scale production of high-quality polymers or compounds obtained from biomass waste.

## Figures and Tables

**Figure 1 polymers-13-02430-f001:**
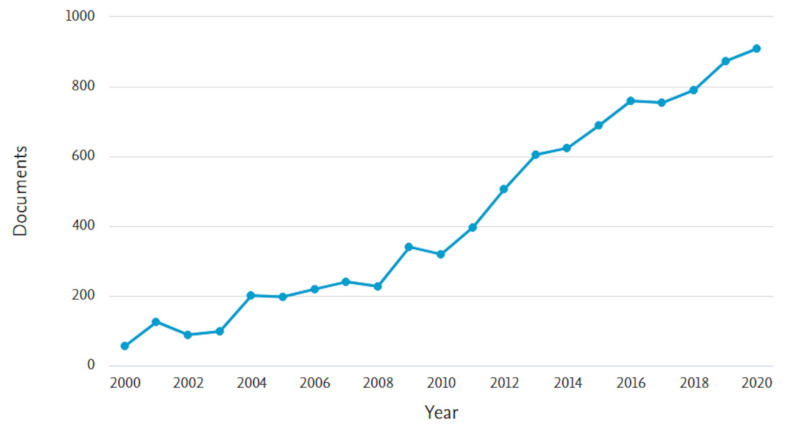
Number of papers per year with topic of molecularly imprinted polymers (Scopus platform, from 2000 to 2020).

**Figure 2 polymers-13-02430-f002:**
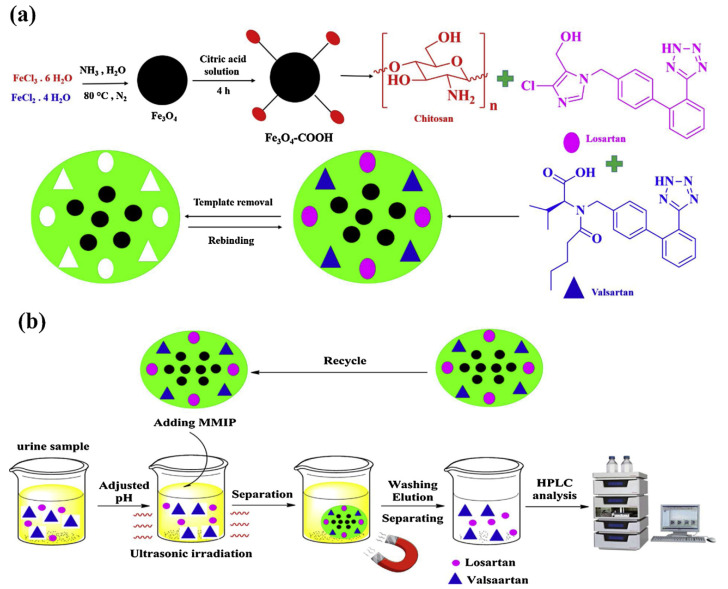
Synthesis procedure of magnetic MIP (**a**) and magnetic MIP for valsartan and losartan extraction (**b**). Reproduced from the work in [49] with permission from Elsevier.

**Figure 3 polymers-13-02430-f003:**
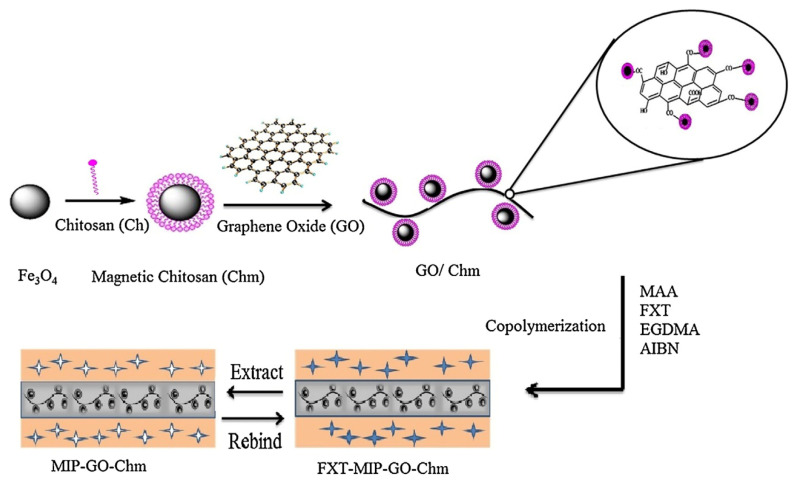
Preparation of magnetic chitosan/graphene oxide. Reproduced from the work in [50] with permission from Elsevier.

**Figure 4 polymers-13-02430-f004:**
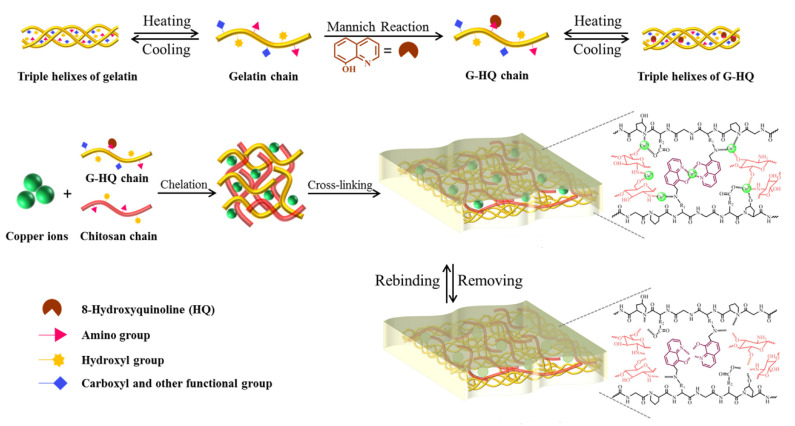
Description of preparation process for gelatin-8-hydroxyquinoline-chitosan (G-HQ-CS) ion imprinted polymers (IIPs). Reproduced from the work in [52] with permission from Elsevier.

**Figure 5 polymers-13-02430-f005:**
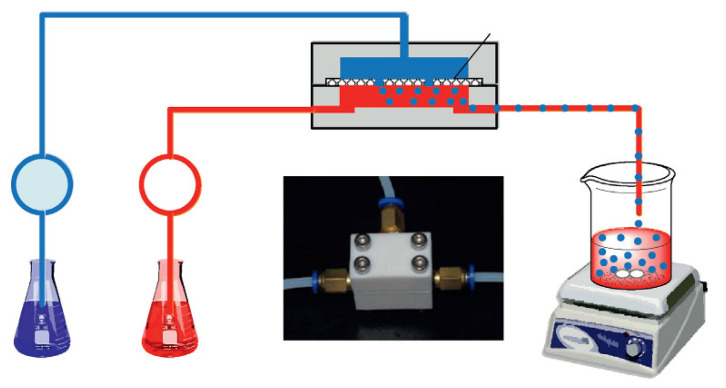
Description of Ca(II)–CS microsphere preparation. Reproduced from the work in [53] with permission from Elsevier.

**Figure 6 polymers-13-02430-f006:**
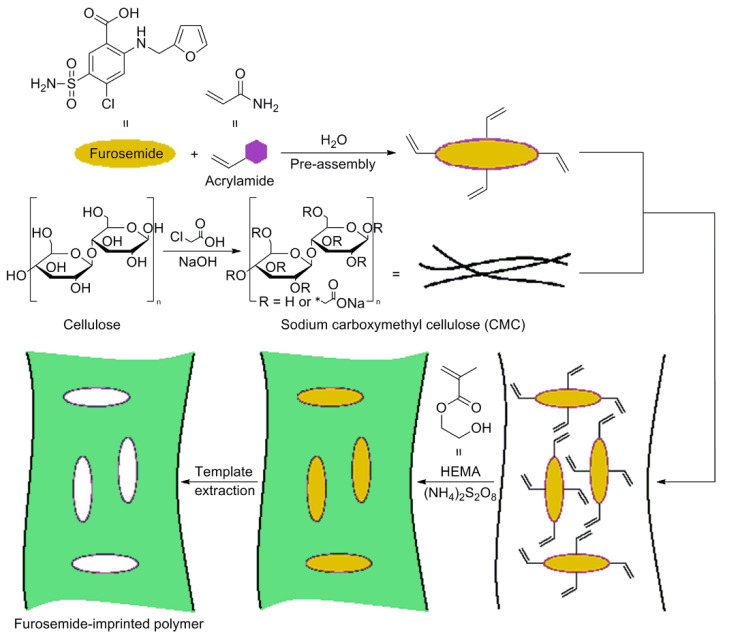
Synthesis procedure of cellulose-based MIP specific for Furosemide drug. Reproduced from the work in [70] with permission from Wiley.

**Figure 7 polymers-13-02430-f007:**
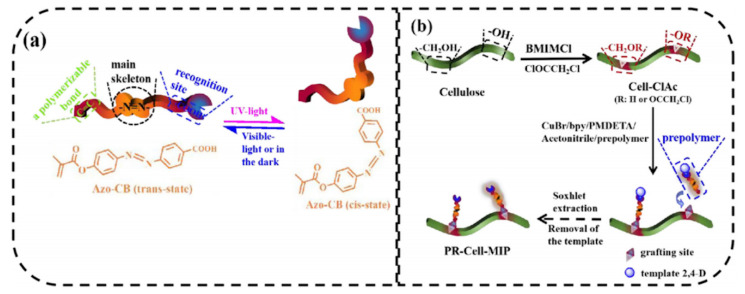
Schematic illustrations for (**a**) photo-responsive properties of the Azo-CB functional monomer and (**b**) synthesis route of PR-Cell-MIP. Reproduced from the work in [73] with permission from Elsevier.

**Figure 8 polymers-13-02430-f008:**
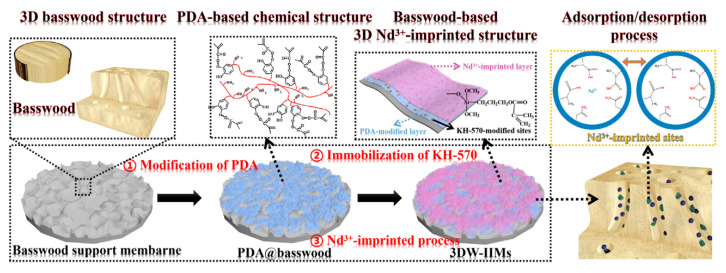
Synthesis route for 3DW-IIMs preparation. Reproduced from the work in [82] with permission from Elsevier.

**Figure 9 polymers-13-02430-f009:**
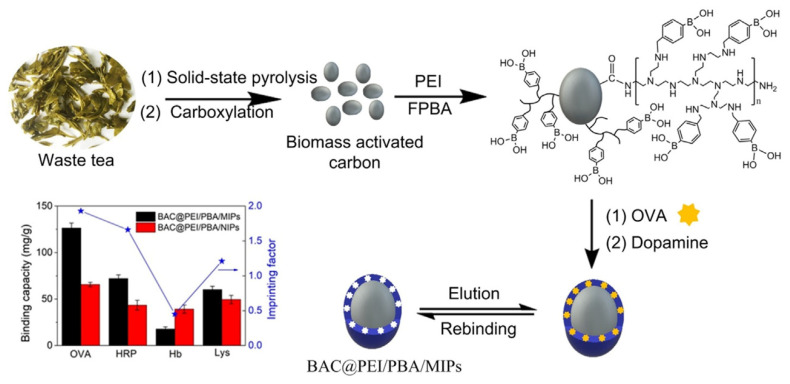
Synthesis procedure, binding capacity and imprinting factors of BAC-based molecularly imprinted polymers. Reproduced from the work in [94] with permission from Elsevier.

**Figure 10 polymers-13-02430-f010:**
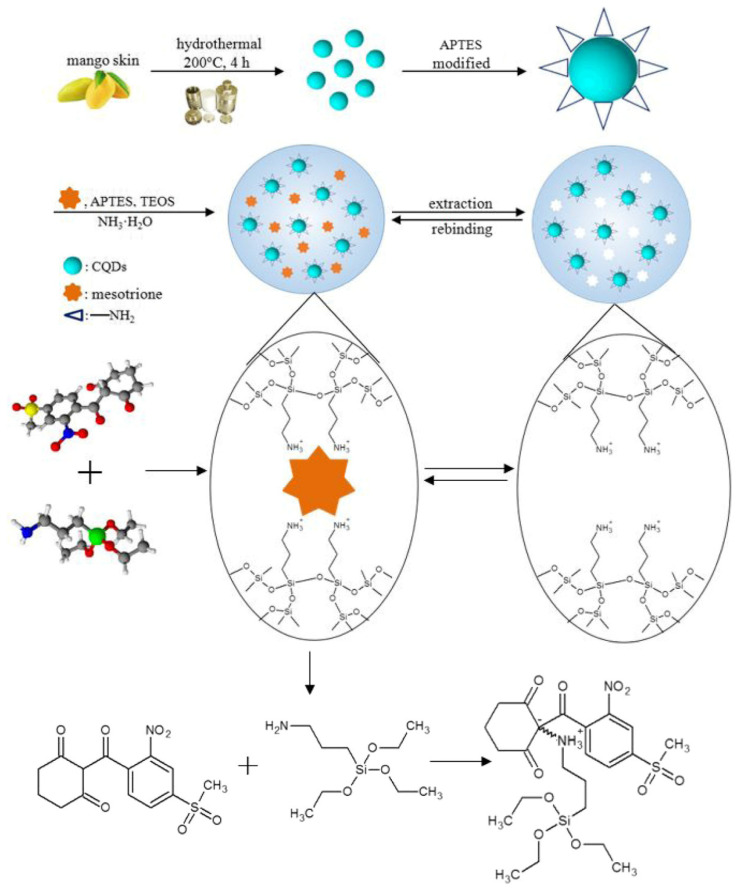
Synthesis preparation of CQDs@MIPs. Reproduced from the work in [108] with permission from Springer Nature.

**Figure 11 polymers-13-02430-f011:**
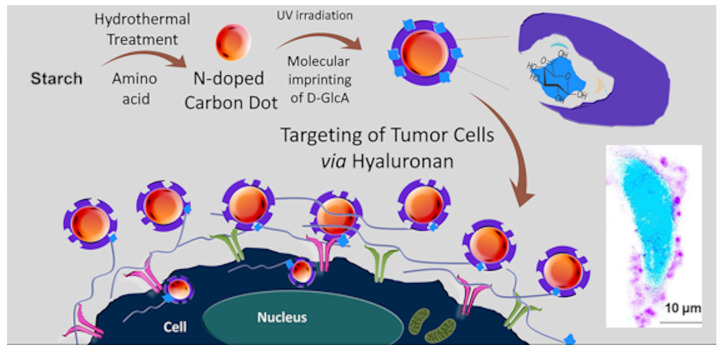
Preparation of core–shell CD-MIPs for bioimaging of cancer cells. Reprinted (adapted) with permission from Demir, B.; Lemberger, M.M.; Panagiotopoulou, M.; Medina Rangel, P.X.; Timur, S.; Hirsch, T.; Tse Sum Bui, B; Wegener, J.; Haupt, K. *ACS Appl. Mater. Interfaces* **2018**, 10, 3305−3313 [110]. Copyright 2018 American Chemical Society.

**Figure 12 polymers-13-02430-f012:**
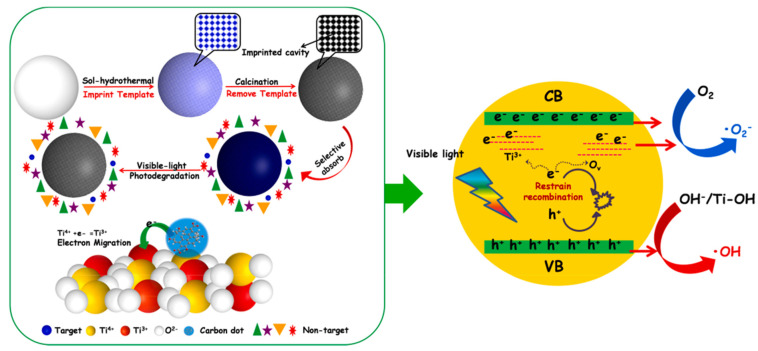
Schematic description of electron transition and photocatalytic process of MIP-TiO_2−x_/CDs under visible light. Reproduced from the work in [26] with permission from Elsevier.

**Figure 13 polymers-13-02430-f013:**
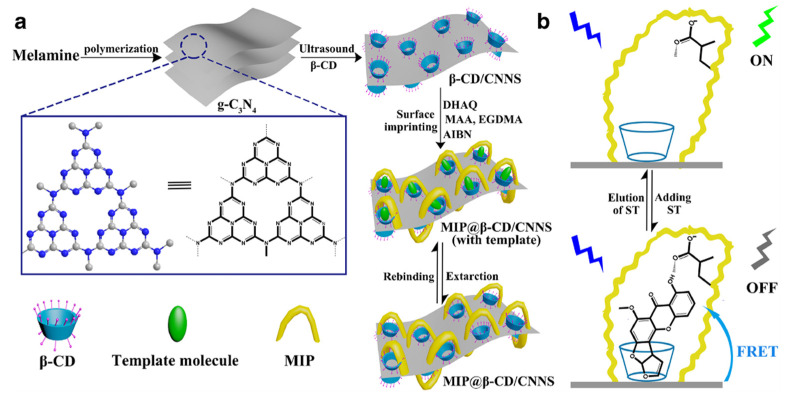
Schematic representation of the synthesis of MIP@β-CD-modified carbon nitride nanosheets (CNNS) (**a**); and its application (**b**). Reproduced from the work in [134] with permission from Springer Nature.

**Figure 14 polymers-13-02430-f014:**
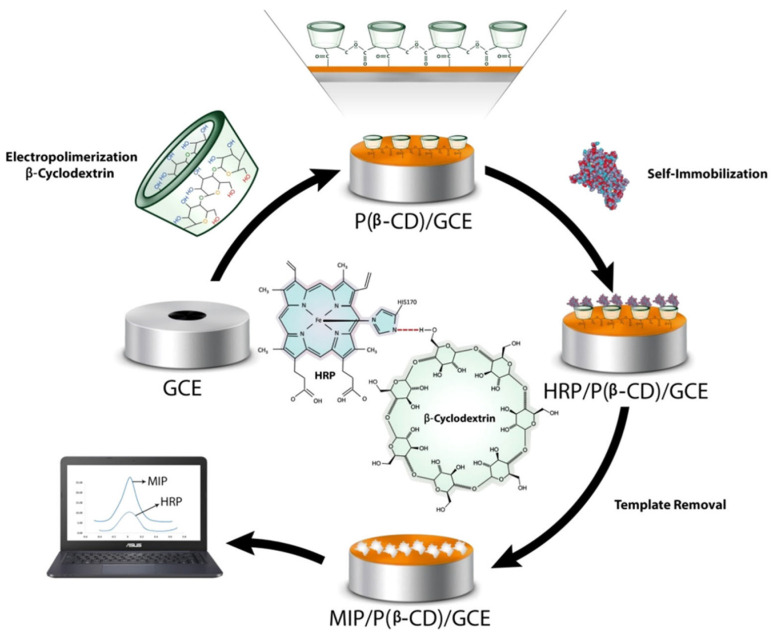
MIP/P(β-CD)/GCE preparation and detection of HRP. Reproduced from the work in [136] with permission from Wiley.

**Figure 15 polymers-13-02430-f015:**
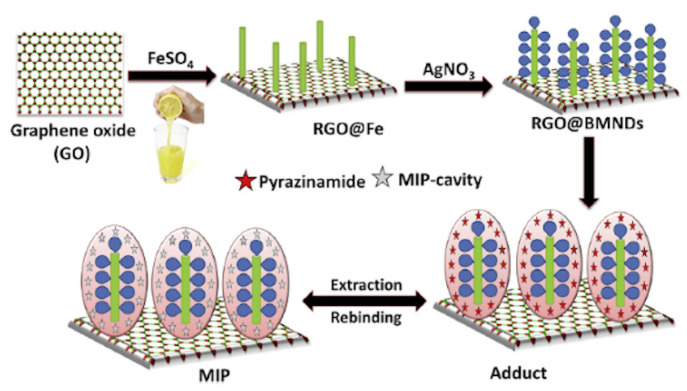
Graphical representation for the synthesis of imprinted reduced graphene oxide@bimetallic nanodendrites. Reproduced from [151] with permission from Elsevier.

**Table 1 polymers-13-02430-t001:** An overview of the reported studies on the use of various waste sources in MIPs preparation.

Waste Material	MIP Composition	MIP Strategy	Target Analyte	Application	Ref.
Chitosan	Core–shell CS-based magnetic MIPs	Surface imprinting	Acrylamide	Clean-up and pre-concentration in biscuit samples	[48]
Chitosan	Core–shell CS-based magnetic MIPs	Surface imprinting	Valsartan Losartan	Simultaneous pre-concentration and determination from urine samples	[49]
Chitosan	MIP on the surface of magnetic CS/graphene oxide	Surface imprinting	Fluoxetine	Separation and pre-concentration in pharmaceutical formulation, human urine and environmental water sample	[50]
Chitosan	Core–shell chitosan gold nanoparticles/decorated MIP	Surface imprinting	Ciprofloxacin antibiotic	Electrochemical biomimetic sensor	[51]
Chitosan	IIPs membrane of gelatin/8-hydroxyquinoline/CS	Crosslinking of gelatin, 8-hydroxyquinoline and CS	Cu(II)	Removal from aqueous solution	[52]
Chitosan	Ion imprinted CS microspheres	Microfluidic technique	Ca(II)	Removal of Cu(II), Cd(II) and Pb(II) from wastewater	[53]
Chitosan	Ion imprinted thiourea modified CS microspheres	Microfluidic technique	Cu(II)	Removal from wastewater	[54]
Cellulose	Water compatible CMC/AA/2-hydroxyethylmethacrylate	Crosslinking of CMC, AA, 2-hydroxyethylmethacrylate	Furosemide	Slow release in drug delivery	[70]
Cellulose	Polypyrrole/Sulphur-CMC IIPs	Glassy carbon electrode modified by electro-polymerizing of pyrrole and Sulphur-IIPs drop coated	Hg(II)	Electrode modifier for the electrochemical detection of Hg(II)	[77]
Cellulose	MIP/CMC coupled with magnetic material	Surface imprinting on magnetic CMC nanocrystals	Fluoroquinolones	Adsorption and determination in water	[56]
Cellulose	Photo-responsive cellulose-based imprinted polymer	Surface-initiated atom transfer radical polymerization	2,4-dichlorophenoxyacetic acid	Photo responsive sorbent material	[73]
Cotton wool cellulose	Hierarchical silica-based ion imprinted mesoporous polymers	Via dual template method	Cd(II) and Pb(II)	Micro solid phase simultaneous extraction from river water and fish muscles	[71]
Rice straw cellulose	Straw-supported IIPs	Surface imprinting combined with AGET-ATRP	La(III)	Sorbent material	[85]
Basswood cellulose	3D-macroporous wood-based membrane decorated with imprinted domains	Two-step-temperature free radical polymerization	Nd(III)	Sorbent material	[82]
Agricultural waste biochar	Activated biochar functionalized with 3-mercaptopropyltrimethoxysilane-based IIPs	Surface imprinting	Cd(II)	Selective removal from wastewater	[91]
Waste tea derived carbon	Activated carbon support of dopamine-based imprinted polymer with multi-boronic acid sites	Surface imprinting	Albumin	Selective capture of glycoprotein	[94]
Sunflower heads	Activated carbon support of acrylamide-based imprinted polymer	Surface imprinting	CO_2_	Selective adsorption in the presence of H_2_O and in simulated flue gas	[92]
Teak ligno-cellulosic material	Activated carbon/MIP/parafin	By mixing paraffin, MIP and activated carbon	Melamine	Electrochemical biosensor	[93]
Sweet potato peels	MIP-coated CDs	Sol–gel polymerization	Oxytetracycline	Fluorescence probe for specific recognition and sensitive detection in honey	[105]
Longan peels	CDs coupled with restricted access materials and MIPs	Multifunctional composites by a simple polymerization method	Metronidazole	Fluorescence probe for specific recognition and sensitive detection in serum	[107]
Mango peels	CDs encapsulated into MIPs	Sol-gel polymerization	Mesotrione	Fluorescent biosensor for detection in corn	[108]
Rosemary leaves	CDs embedded in silica MIPs	Reverse microemulsion and surface imprinting	Thiabendazole	Optical probe for quantification in apple, orange, and tomato juices	[109]
Starch	MIPs coated CDs	Epitope approach with glucuronic acid	Hyaluronan	Cancer cell biotargeting and bioimaging	[110]
Lignite	MIP-TiO_2-x_/CDs nanocomposite	Two-step hydrothermal calcination method	Methylene blue	Photocatalytic degradation in wastewater	[26]
Cyclodextrins	Magnetic MIP of MAA-βCD	Bulk polymerization	Bisphenol A	Magnetic sorbent for pollutant removal from water	[117]
Cyclodextrins	β-CD/functionalized carbon nitride nanosheets	Surface imprinting	Sterigmatomycin	Fluorometric detection	[134]
Cyclodextrins	β-CD/graphitic carbon nitride composite	β-CD non-covalently functionalized carbon nitride	Platelet derived growth factor BB	Electrochemiluminescent aptasensor	[135]
Cyclodextrins	poly ß-CDs on the electrode surface	Electropolymerization	horseradish peroxidase and H_2_O_2_	Electrochemical biosensor in human plasma	[136]
Eucalyptus extract	Magnetic MIP-green iron nanoparticles using eucalyptus extract	Surface imprinting	Non-steroid anti-inflammatory drugs	Magnetic sorbent material	[150]
Lemon juice	Imprinted reduced graphene oxide@bimetallic nanodendrites	Surface imprinting	Pyrazinamide	Electrochemical sensor	[151]
Guava Leaves aqueous extract	Silver imprinted zinc nanoparticles	Plant extracts mediated synthesis	Methylene blue	Photocatalytic water detoxification	[152]
Soybean oil	MIP using epoxidized soybean oil as crosslinking agent	Thermal polymerization	Resveratrol	Biopesticide delivery system	[153]
Solanum tuberosum starch	Ion imprinted Starch/PVA nanofibers	Electro-spinning	Thorium (IV)	Sorbent material	[154]

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
