# Peer review of "Green Aspects in Molecularly Imprinted Polymers by Biomass Waste Utilization"

_polymers, 2021, doi:10.3390/polym13152430_

Round 1
Reviewer 1 Report
I believe that the authors of review articles should choose the content of that review, not reviewers. This particular review could be more critical but it certainly is comprehensive. My comments should be viewed as suggestions that might improve the presentation rather than requirements. The only action that I view as essential is editing to put this manuscript in idiomatic English. Even with this, the writing is mostly fine but there are a few places where changes would definitely help.
The authors use the acronym, MIT, for molecularly imprinting technology. This makes sense in that they cover situations where green chemistry is used to prepare components of molecularly imprinted polymers rather than actually preparing imprinted polymers. However, I think they would be better off referring to MIP technology rather than using a new acronym that won’t be familiar to readers.
I think an introduction that says more about the contents of this review and acknowledges that this review is noncritical would help readers know what they can and can not get from this review.
I also think an introduction that puts this research into context and points the way to potential practical applications would be valuable for readers like myself who are not intimately familiar with all aspects of molecularly imprinted polymer technology. This introduction might also deal with the issue of scale. Purification applications may require a lot of material and thus make a dent in the amount of available waste product. However, most analytical applications do not require that much material.
Author Response
We wish to thank you for the evaluation and comments about our manuscript.
We have modified the manuscript taking into account your comments/suggestions.
All changes have been written in track changes mode in the manuscript.
As suggested from the reviewer we have attempted to improve the English language and corrected some grammatical errors, written in track changes mode in the text. For instance:
- Page 1, “Aim” instead of “spirit”
- Page 1, “almost thirty years later the introduction of green chemistry principles… governments, and scientific societies all over the world” were removed
- Page 1, “In the light of these considerations, one of the main objectives is to drastically reduce the environmental impact of waste with the scope to promote the concept for their recycling and transforming it into value added products.” instead of “As a consequence it is fundamental, in view of reducing the environmental impact of wastes, to change the idea of waste moving from pollutant to secondary renewable resource.”
- Pag 1, “however, it is important to remark that, in some cases, waste biomass-assisted synthesis is less costly, environmentally friendly and renewable strategy, and therefore, wastes may become ideal renewable resources for production of functionally engineered macro or nanomaterials.” Has been added
- Pag 2, “have be also considered” instead of “can be also considered”
- 4 “Fibers” instead of “Fibres”
- 4 “contexts” instead of “contests”
- 4, “electrical” instead of “electrica”
- Pag 14, “biomass” instead of Biomasse
- 19, “In particular, it was underlined” instead of “underlining”
- Pag 19, “availability” instead of “disponibility”…
- …
Please find below our answers to your comments:
- As suggested from the reviewer we changed the acronym MIT with MIP technology firstly in the Abstract. Moreover into the manuscript was changed at pag. 7, 14, 17, 19.
- In the introduction, in order to say more about the contents of this review and help readers know what they can and can not get from this review we added at the end of introduction the following sentence “The aim of this review is to give to the reader an overview of recent works that have seen the use of biomass in the preparation of MIPs. The next paragraph has been divided into six subparagraphs each of them dealing with a specific biomass waste. In detail, the following wastes will be taken into consideration: chitosan, cellulose, activated carbon, carbon dots, cyclodextrins and waste extracts. The herein paper is not a critical collection of results of the above-cited topics but a comprehensive and useful summary of the state-of-art as starting point for future development in the topic of biomass waste for MIPs preparation.”
- Following the reviewer suggestion to point the applications of MIPs we deepened application discussion into introduction section (pag. 2) by adding the following sentences “It is worth noting that several different MIPs formats have been developed such as bulk or monoliths, microspheres and core-shell materials, magnetically susceptible and stir-bar imprinted materials which are applicable as sorbents of solid-phase extraction. MIP sorbents are capable to determine target analytes and ions in a very complex environment such as blood, urine, soil, or food and it have gained interest in trace analysis of pollutants in environmental samples. On the other side, MIPs in sensor field have seen electrochemical sensing as the preferred analytical technique, followed by optical detection with high improvements in sensitivity as well as a shift in research and development toward real-life applications and point-of-care testing in real human samples. Moreover, the research in MIP sensor devices is nowadays starting moving from laboratory research to large-scale manufacturing [22].”
- The reviewer take into consideration also the possibility to discuss on the issue of scale in the introduction section. However we guess that this issue goes beyond the aim of this review because we want only describe how waste have been used in MIP preparation emphasizing the potentiality of this field which is yet in an early stage. Certainly in the future the issue scale should be also evaluated.
Corresponding author

Reviewer 2 Report
The manuscript “Green aspects in molecularly imprinted polymers by biomass waste utilization” by Del Sole et al. reviews the recent progress in the use of biomass waste for imprinted polymers preparation. Although, authors had described most of materials, some sentences may not very clear. Therefore, I would suggest authors may take at least a major revision before publication. Here are the comments and suggestions: 1. Some reports mentioned the purification of chitosan is not a green process; therefore, the title of this manuscript has to revised. 2. The abstract did not describe the content of this work. 3. There are a lot of grammar typos in this manuscript. 4. Additional tables summarized the materials mentioned in this work are suggested to add. 5. The last two paragraphs of conclusions seem more like an introduction of this work.Author Response
Reply to reviewer 2
REVIEWER 2
We wish to thank you for the evaluation and comments about our manuscript.
We have modified the manuscript taking into account your comments/suggestions.
All changes have been written in track changes mode in the manuscript.
Please find below our answers to your comments:
- The reviewer observed that some reports mentioned for purification of chitosan are not a green process and he suggested revising the title of this manuscript. However, we guess that the green aspects are important in various points of this review thus we preferred do not change the title but add an explanation sentence into the Chitosan paragraph (see pag. 4) “Even if chitosan satisfies green principles because it is a widespread and cheap fish biomass waste, the green principles are not always applied in the whole process that use chitosan. Thus, in a more comprehensive green perspective it is desirable that researchers will keeps in mind and will evaluates the green principles on all the design steps.”
- Following the proper observation of the reviewer on the abstract, we slightly modified the abstract in order to better describe the content of this review work but taking into account also the limit of words number (200). Thus the final part of the abstract was changed as follow: “…which can be addressed to the production of high value carbon-based materials with different applications.
This review aims to focus and explore in detail the recent progress in the use of biomass waste for imprinted polymers preparation. Specifically, different types of biomass waste in MIP preparation will be exploited: chitosan, cellulose, activated carbon, carbon dots, cyclodextrins and waste extracts, describing the approaches used in the synthesis of MIPs combined with biomass waste derivatives.”
3. As suggested from the reviewer we have attempted to improve the English language and corrected some grammatical errors, written in track changes mode in the text. For instance:
-Page 1, “Aim” instead of “spirit”
- Page 1, “almost thirty years later the introduction of green chemistry principles… governments, and scientific societies all over the world” were removed
- Page 1, “In the light of these considerations, one of the main objectives is to drastically reduce the environmental impact of waste with the scope to promote the concept for their recycling and transforming it into value added products.” instead of “As a consequence it is fundamental, in view of reducing the environmental impact of wastes, to change the idea of waste moving from pollutant to secondary renewable resource.”
- Pag 1, “however, it is important to remark that, in some cases, waste biomass-assisted synthesis is less costly, environmentally friendly and renewable strategy, and therefore, wastes may become ideal renewable resources for production of functionally engineered macro or nanomaterials.” Has been added
- Pag 2, “have be also considered” instead of “can be also considered”
- 4 “Fibers” instead of “Fibres”
- 4 “contexts” instead of “contests”
- 4, “electrical” instead of “electrica”
- Pag 14, “biomass” instead of Biomasse
- 19, “In particular, it was underlined” instead of “underlining”
- Pag 19, “availability” instead of “disponibility”….
- …
4. As correctly observed from the reviewer, additional tables summarizing the materials mentioned in this work should be added. To this aim, we added a Table (Table 1) at pag. 19 that summarize all the works mentioned in this review highlighting the materials used. Moreover a comment on the Table was added in the text (pag. 19) into a new specific paragraph (2.7 Summary) .
5. The reviewer observed that the last two paragraphs of conclusions seem more like an introduction of this work. We improved the conclusion section adding the following sentences at the end of this section: “Although numerous examples of preparation of MIPs by using different biomass waste have been recently described in literature, many challenges still exist. First of all, the use of biomass waste in combination with other materials, such as nanoparticles or magnetic materials, represent a great potential since it allows to obtain novel composite materials with unique properties. However, it requires applying improvement in the green concept to the whole design process. Second, it is still a big challenge to realize the large-scale production of high-quality polymers or compounds obtained from biomass waste.”
Corresponding author
